# FinaleMe: Predicting DNA methylation by the fragmentation patterns of plasma cell-free DNA

Yaping Liu [1,2,3,4,5,6,7,8,9] ✉, Sarah C. Reed[8,10], Christopher Lo[8], Atish D. Choudhury [8,11], Heather A. Parsons [11], Daniel G. Stover [11], Gavin Ha [8], Gregory Gydush[8], Justin Rhoades[8], Denisse Rotem[8], Samuel Freeman [8], David W. Katz [1,2,3], Ravi Bandaru [1,2,3], Haizi Zheng [3], Hailu Fu [1,2,3], Viktor A. Adalsteinsson [8] ✉ & Manolis Kellis [8,9] ✉

Analysis of DNA methylation in cell-free DNA reveals clinically relevant biomarkers but requires specialized protocols such as whole-genome bisulfite sequencing. Meanwhile, millions of cell-free DNA samples are being profiled by whole-genome sequencing. Here, we develop FinaleMe, a non-homogeneous Hidden Markov Model, to predict DNA methylation of cell-free DNA and, therefore, tissues-of-origin, directly from plasma whole-genome sequencing. We validate the performance with 80 pairs of deep and shallow-coverage whole-genome sequencing and whole-genome bisulfite sequencing data.

DNA methylation plays an instrumental role in gene regulation during disease progression and embryonic development[1,2]. Genome-wide DNA methylation level in cell-free DNA (cfDNA) has been extensively studied for disease diagnosis and prognosis[3–7]. The current gold standard to measure DNA methylation from cfDNA molecules is bisulfite sequencing[8]. However, sodium bisulfite treatment causes non-uniform sequence-dependent degradation of most DNA fragments[9,10]. The substantial loss of input DNA during the bisulfite treatment limits the sensitivity of diagnostic tests and analyses[11]. Recent advances in enzymatic conversion and long-read sequencing approaches have partly mitigated these issues but require specialized protocols[12–16].

Unlike genomic DNA (gDNA), cfDNA is not randomly fragmented and its fragmentation pattern is highly associated with the local epigenetic background[17,18]. Several recent studies have identified significantly different DNA fragmentation patterns between methylated and unmethylated cfDNA molecules[7,19,20]. These findings suggest the

possibility of computationally inferring DNA methylation levels from cfDNA fragmentation patterns. One recent study provided a proof-of-concept solution to predict the binary status of DNA methylation in high-coverage whole-genome bisulfite sequencing (WGBS) through a deep-learning model[19]. However, the ability to predict methylation status from cfDNA whole-genome sequencing (WGS) remains unexplored. The 2020 American College of Obstetricians and Gynecologists (ACOG) guidelines recommend non-invasive prenatal testing (NIPT) for all pregnancies regardless of risk, which will eventually result in millions of shallow-coverage (~0.1X-1X) cfDNA WGS every year in the US. In addition, hundreds of thousands of cfDNA WGS samples have already been sequenced for cancer early detection and other purposes worldwide by academic communities and commercial entities[21].

Here, to leverage cfDNA WGS datasets and advance understanding of gene regulation and human health[22], we develop a computational method, named FinaleMe (**F**ragmentat**I**o**N A**na**L**ysis of cEll-

[1]Department of Biochemistry and Molecular Genetics, Feinberg School of Medicine, Northwestern University, Chicago, IL 60611, USA. [2]Robert H. Lurie Comprehensive Cancer Center of Northwestern University, Chicago, IL 60611, USA. [3]Division of Human Genetics, Cincinnati Children's Hospital Medical Center, Cincinnati, OH 45229, USA. [4]Division of Biomedical Informatics, Cincinnati Children's Hospital Medical Center, Cincinnati, OH 45229, USA. [5]Department of Pediatrics, University of Cincinnati College of Medicine, Cincinnati, OH 45229, USA. [6]University of Cincinnati Center for Environmental Genetics, Cincinnati, OH 45229, USA. [7]University of Cincinnati Cancer Center, Cincinnati, OH 45229, USA. [8]Broad Institute of MIT and Harvard, Cambridge, MA 02142, USA. [9]Massachusetts Institute of Technology, Computer Science and Artificial Intelligence Laboratory, Cambridge, MA 02139, USA. [10]Medical Scientist Training Program, Vanderbilt University School of Medicine, Nashville, TN, USA. [11]Dana-Farber Cancer Institute, Boston, MA, USA. ✉e-mail: lyping1986@gmail.com; viktor@broadinstitute.org; manoli@mit.edu

free DNA **Me**thylation), to predict the DNA methylation status in each CpG at each cfDNA fragment and obtain the continuous DNA methylation level at CpG sites, mostly accurate in CpG rich regions. We further predict the associated tissues-of-origin status from the inferred methylation patterns. We validate the predictions of both methylation level and tissues-of-origin status using paired WGS and WGBS of plasma cfDNA from the same tube of blood across different physiological conditions at deep (-16-39X) and shallow (-0.1X) WGS.

## Results

Since DNA methylation has been tightly correlated with nucleosome occupancy[23,24], we hypothesized that if the boundaries of cfDNA fragments are biased by their association with nucleosomes, then the fragmentation pattern observed in each cfDNA molecule should indicate its associated DNA methylation pattern and thus its tissue-of-origin. To evaluate this hypothesis, we first studied the correlation between fragment size and mean methylation level of DNA fragments from publicly available WGBS of cfDNA and gDNA of buffy coat samples from two healthy individuals[7] (Fig. 1). Replicate samples of cfDNA showed waved methylation patterns at mono-nucleosomal lengths that were not present in the gDNA samples. This observation supported the hypothesis that the fragmentation pattern of cfDNA can provide information related to the DNA methylation level.

Next, we built a non-homogeneous Hidden Markov Model (HMM), named FinaleMe, to predict the methylation status in cfDNA (details in Methods and Supplementary Methods, Fig. 2). Since CpGs are not evenly distributed in the human genome, we incorporated the distance between CpG sites into the model and utilized the following three features: fragment length, normalized coverage, and the distance of each CpG to the center of the DNA fragment (Fig. 1b). We first

evaluated the model using high-coverage WGBS of cfDNA (from non-pregnant healthy individuals), masking the methylation status, and then benchmarked the model performance using the ground truth DNA methylation states at each CpG in each DNA fragment. After sampling an equal number of the methylated and unmethylated CpGs, we observed high performance in predicting the methylation status at each single CpG from each DNA fragment based on the area under the receiver operating characteristic curve (auROC) within CpG-rich regions (auROC=0.91, for CpGs at fragments with ≥5 CpGs, Fig. 1c).

To further benchmark the model performance in cfDNA WGS, we generated our own matched high-coverage WGS (-16–39X) and WGBS (-10–15X) data from plasma cfDNA samples within the same tube of blood in healthy individuals and a prostate cancer patient (Fig. 3a, Supplementary Data 1–3). Without using cfDNA WGBS data, we trained the HMM model and predicted the methylation level from the same cfDNA WGS dataset. By comparing the results with the methylation level at CpG sites in the reference genome from matched WGBS, we achieved a high correlation at single-CpGs and 1 kb windows in CpG-rich regions (CpG island and CpG island shore regions, Fig. 3b, c). At differentially methylated regions (DMRs) detected in the cfDNA WGBS between cancer and healthy individuals at CpG-rich regions, we also observed consistent methylation changes in the predicted methylation levels from matched cfDNA WGS (Fig. 3d). To check the potential overfitting problem of the model, we further trained and decoded the model for gDNA WGS from cancer and normal blood cells, in which the fragments are sonicated and do not have a correlation with the epigenetics status. The predicted results for gDNA WGS did not show any methylation differences between cancer and normal cells in the DMRs detected at the matched gDNA WGBS datasets (Supplementary Fig. 1a). This result suggested that the differential methylation we

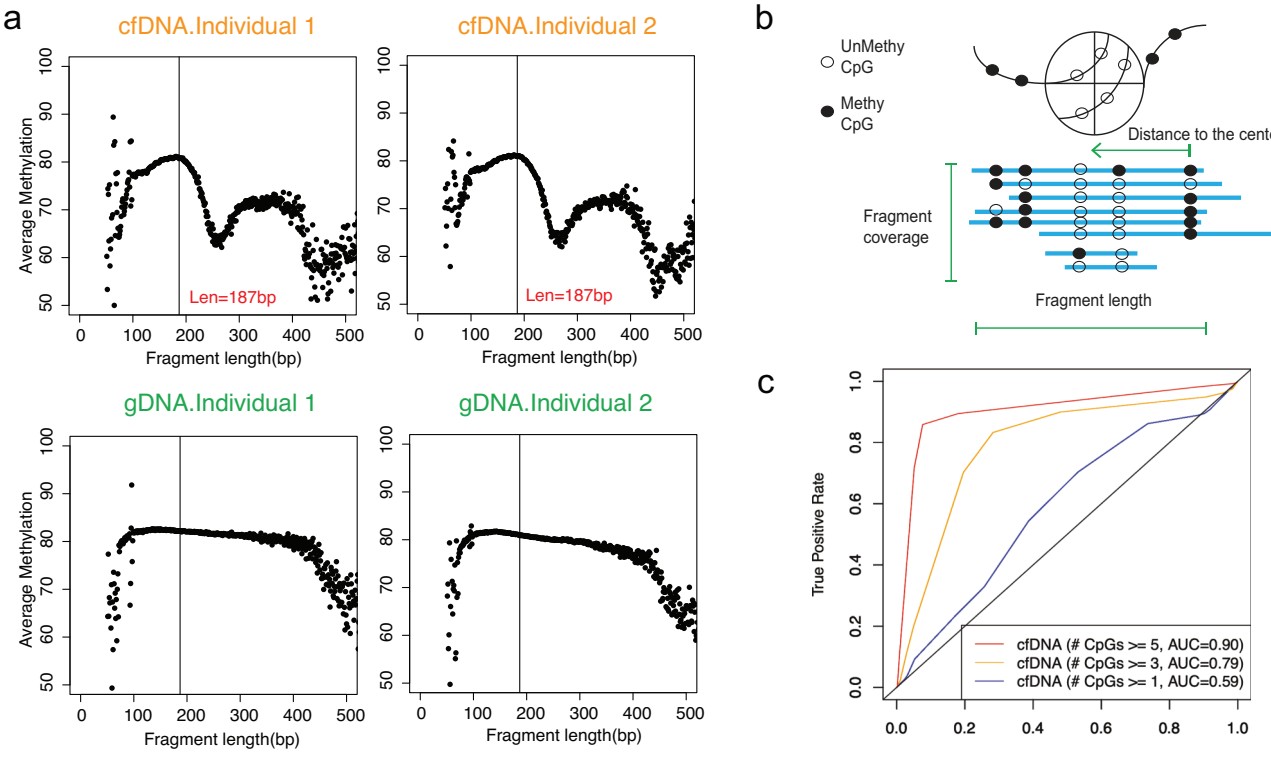

**Fig. 1 | Inferring DNA methylation from high-coverage whole-genome bisulfite sequencing. a** The correlation between mean DNA methylation and fragment lengths in cfDNA and gDNA WGBS in healthy individuals. **b** Diagram of the features utilized for the inference of DNA methylation level. Unmethy: unmethylated CpG. Methy: methylated CpG. **c** Receiver operating characteristic curve (ROC) for the model performance at deep WGBS in fragments with different numbers of CpGs. AUC: Area under the ROC Curve. Red line represents the ROC for fragments with equal or more than 5 CpGs. Yellow line represents the ROC for fragments with equal or more than 3 CpGs. Blue line represents the ROC for fragments with equal or more than 1 CpG. Source data are provided as a Source Data file.

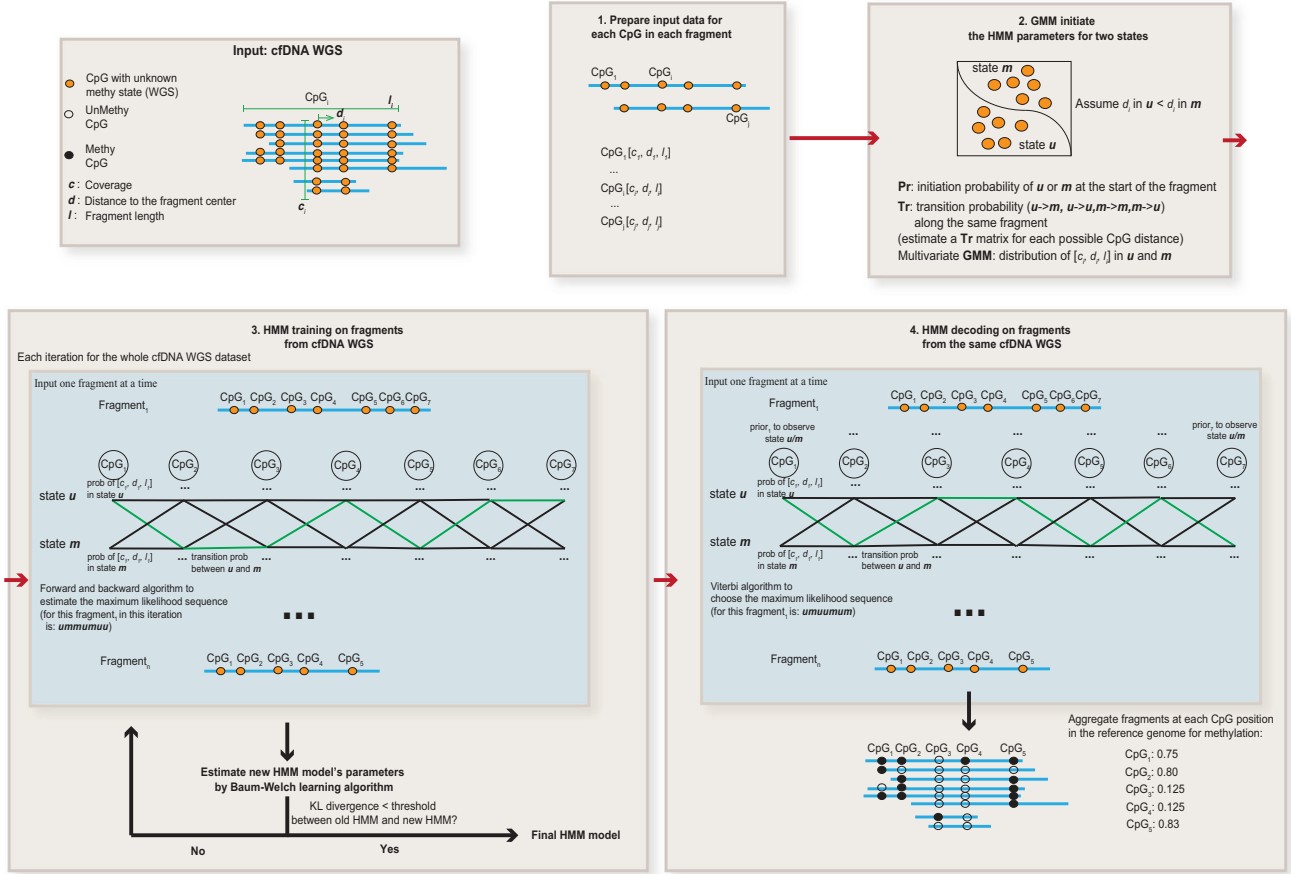

**Fig. 2 | Summary of the HMM model to infer DNA methylation status.** GMM Gaussian Mixture Model, HMM Hidden Markov Model, KL divergence Kullback–Leibler distance.

predicted in cfDNA WGS was not driven by the methylation prior we used but indeed the fragmentation features. However, we noticed that, in the CpG-poor regions, FinaleMe did not work as well as in CpG-rich regions (Supplementary Fig. 1b). We further assessed the methylation level at important regulatory elements, such as CpG island (CGI) promoters (Fig. 3e), 5′exon boundaries, and CTCF motifs (Supplementary Fig. 2). These results showed a high correlation between the ground truth (WGBS) and the prediction (WGS) in cfDNA from both healthy individuals and the cancer patient (Fig. 3e, Supplementary Figs. 2, 3), but not in gDNA dataset (Supplementary Fig. 4).

Since DNA methylation in CGI and CGI shore regions are often cell-type-specific, we further estimated the tissue-of-origin in cfDNA by using DNA methylation levels that were measured or predicted using WGBS and WGS, respectively. We found similar tissue-of-origin profiles between predicted and measured methylation levels for each of the individuals in both cancer and healthy conditions (Fig. 3f), which was also largely consistent with other previous tissues-of-origin studies by cfDNA WGBS[3,6].

Deep coverage WGBS and WGS remain costly for routine clinical application. Many publicly available cfDNA WGS datasets are sequenced with shallow coverage (0.1–1X). We sought to determine whether we could predict DNA methylation levels using ultra-low-pass whole-genome sequencing (~0.1X, ULP-WGS). We generated matched ULP-WGS and ultra-low-pass WGBS (~0.1X, ULP-WGBS) of cfDNA from 77 individuals, including healthy donors, breast, and prostate cancer patients (Supplementary Data 1–3). We examined the methylation level globally and at important regulatory elements, such as CGI promoters, and observed similar average methylation profiles in predicted and

measured methylation levels from ULP-WGS and WGBS, respectively (Fig. 4a, b). We also observed the differential methylation level in ULP-WGS at differentially methylated regions detected in ULP-WGBS (Supplementary Fig. 5). Next, we assessed whether methylation levels from ultra-low-pass sequencing could be utilized for the estimation of tissues-of-origin. We downsampled the deep coverage sequencing results and found largely consistent tissue-of-origin estimates with ultra-low-pass sequencing (Supplementary Fig. 6). Finally, we estimated the tissue-of-origin in both ULP-WGS and ULP-WGBS. We found consistent results between the two assays. The fractions of prostate or breast-originated cell types are low in healthy individuals and showed a high correlation with tumor fraction as estimated by copy number variations (ichorCNA) across all samples in both assays (Fig. 4c). These results suggested that the application of FinaleMe to ULP-WGS is consistent with ULP-WGBS for both DNA methylation and tissues-of-origin predictions.

## Discussion

Our study demonstrates the ability to infer cfDNA methylation level and tissues-of-origin status directly from deep and shallow-coverage cfDNA WGS. This overcomes a major hurdle associated with bisulfite conversion of limited amounts of cfDNA and, more importantly, enables the usage of a large number of existing, publicly available cfDNA genomic datasets for epigenetic analysis. Our predictions are most accurate in CpG-rich regions of the genome but not in CpG-poor regions. Further work is required to improve the predictions in CpG-poor regions for the detection of other disease-related methylation features, such as the partially methylated domains in cancers.

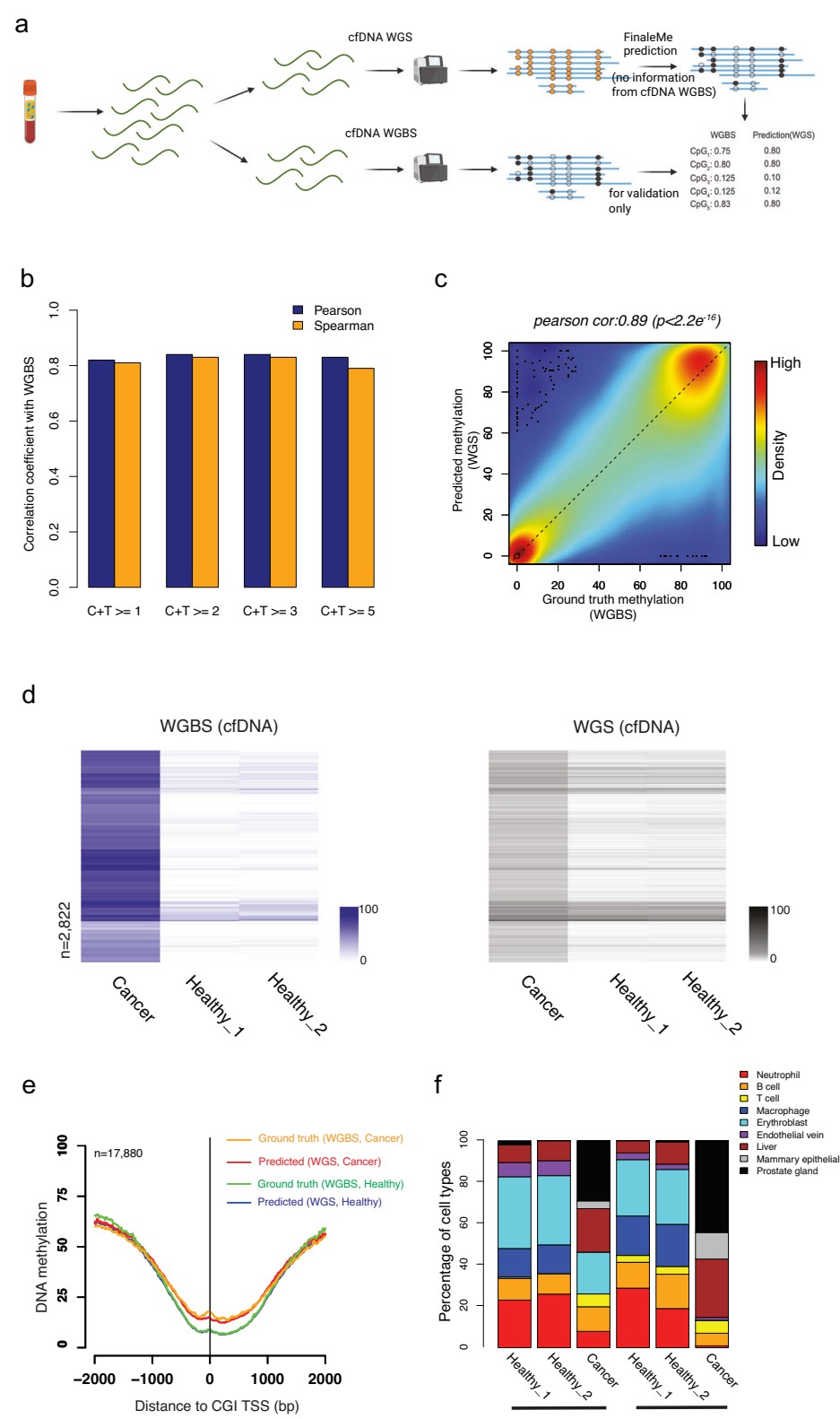

Moreover, the Bayesian prior we utilized from genomic DNA methylome may cause overfitting problems and the false positive call of DMRs in cancer WGS. Previous studies have suggested that analysis of tissue-of-origin is possible based on analysis of nucleosome spacing in WGS of cfDNA[17]. However, only the relative rank of most related cell types is estimated in deep WGS. The tissues-of-origin estimation from inferred DNA methylation here can provide the estimation of absolute fractions in each cell type and utilize the rich reference methylome resources. Although we do not expect to replace bisulfite sequencing for direct measurement of methylation levels, we provide a generalizable method that could enable the methylation analysis of cfDNA samples with limited material or samples that would otherwise only undergo genomic profiling.

**Fig. 3 | Inferring DNA methylation from high-coverage whole-genome sequencing. a** Workflow to benchmark the model performance. Created with BioRender.com. **b** Pearson and Spearman correlation of DNA methylation at single CpGs with different coverages at CpG island and CpG island shore regions between matched cfDNA WGBS and WGS. Blue bars represent the Pearson correlation coefficient. Orange bars represent the Spearman correlation coefficient. **c** Scatterplot of DNA methylation level within 1 kb non-overlapped bins ($n = 116,133$) at CpG island and CpG island shore regions between matched cfDNA WGBS and WGS. The correlation coefficient and *p*-value is calculated by two sided Pearson correlation test in cor.test function in R. **d** Heatmap of measured (left panel, cfDNA WGBS, purple) and predicted (right panel, matched cfDNA WGS, black) DNA methylation level at hypermethylated differentially methylated windows (1 kb)

characterized in CGI and CGI shore regions ($n = 2822$). The row orders in both WGBS and WGS datasets were based on the clustering of DNA methylation levels in WGBS only. **e** Average ground truth (WGBS) and predicted (WGS) DNA methylation level at CpG island promoter regions ($n = 17,880$) from cancer and healthy individuals. Orange line represents the ground truth from WGBS in the cancer patient. Red line represents the predicted value from WGS in the cancer patient. Green line represents the ground truth from WGBS in the healthy individual. Blue line represents the predicted value from WGS in the healthy individual. **f** The fraction of cell types that contributed to cfDNA was estimated by matched WGS and WGBS. Red: Neutrophil. Orange: B cell. Yellow: T cell. Blue: Macrophage. Cyan: Erythroblast. Purple: Endothelia vein. Brown: Liver. Gray: Mammary epithelia. Black: Prostate gland. Source data are provided as a Source Data file.

## Methods

### Ethics approval and consent to participate
This research study was approved by the Broad Institute Institutional Review Board in accordance with the Declaration of Helsinki. De-identified plasma sample collection was approved by the Dana-Farber Cancer Institute and Broad Institute Institutional Review Boards. All participants provided written informed consent to participate.

### Clinical samples
Cancer patient blood samples were obtained from appropriately consented patients as described in Adalsteinsson et al.[25]. Healthy donor blood samples were obtained from appropriately consented individuals from Research Blood Components (http://researchbloodcomponents.com/services.html). Samples were collected and fractionated as described in Adalsteinsson et al.[25].

### Whole-genome bisulfite sequencing of cfDNA
Library construction was performed on 25 ng of cfDNA using the Hyper Prep Kit (Kapa Biosystems) with NEXTFlex Bisulfite-Seq Barcodes (Bioo Scientific) and methylated adapters (IDT) along with HiFi Uracil+ polymerase (Kapa Biosystems) for library amplification. NEXTFlex Bisulfite-Seq Barcodes were used at a final concentration of 7.5 μM and the EZ-96 DNA Methylation-Lightning MagPrep kit (Zymo Research) was used for bisulfite conversion of the adapter-ligated cfDNA prior to library amplification. Libraries were sequenced using paired-end 100 bp in the platform of HiSeq2500 (Illumina) with a 20% spike of PhiX.

### Whole-genome sequencing of cfDNA
Library construction was performed on 5–20 ng of cfDNA using the Hyper Prep Kit (Kapa Biosystems) and custom sequencing adapters (IDT) on a Hamilton STAR-line liquid handling system. Libraries were sequenced using paired-end 100 bp in the platform of the HiSeq2500 (Illumina).

### Model development and training
**Data preprocessing.** For WGS data, reads were aligned to the human genome (GRCh37) using BWA-MEM 0.7.15[26] with default parameters. Each fragment containing CpGs in the autosomal chromosomes reference genome was used for the analysis. Fragment lengths of more than 500 bp or less than 30 bp were discarded. Regions with coverage more than 250× or ENCODE blacklist regions (merged wgEncodeDukeMapabilityRegionsExcludable and wgEncodeDacMapabilityConsensusExcludable) were also discarded. Only high-quality reads were considered in the following analysis (high quality: uniquely mapped, no PCR duplicates, both of ends are mapped with mapping qualities more than 30 and properly paired). To calculate the methylation status for each CpG in each fragment, only bases with a base quality of more than 5 were used.

For cfDNA WGBS data, a recent study demonstrated that the existence of the jagged-end at the end of cfDNA fragment will affect the estimation accuracy of DNA methylation[27]. We first generated the

M-bias plot by using Bismark[28] to map the reads without trimming (see Supplementary Fig. 7). To avoid the artifact potentially brought by the jagged end for Fig. 1a, we trimmed the 40 bp from the 5′ end and 10 bp from 3′ end at the R2 reads. The 3′ end of R1 reads seems to be not affected by the jagged-end problem. However, in CpG islands (often open chromatin regions), cfDNA fragments are usually very small. To avoid the potential bias at these small fragments, we also trimmed 40 bp from 3′ end at the R1 reads, and the results were still largely the same. After trimming, reads were aligned to the human genome (GRCh37) using Bismark (v0.22.3) with bowtie2 (v2.3.5)[29]. The methylation status of CpGs was counted from the first converted cytosine in each of the fragments as described in Bis-SNP[30]. Fragment coverage at each CpG site was first normalized by dividing the total number of high-quality reads in the bam file. Further, the three features (fragment length, normalized coverage, and distance to the center of the fragment) were transformed into Z-score by the mean and standard deviation of the features within the same bam file as the input for the HMM model (Fig. 2). All details are implemented in CpgMultiMetricsStats.java (with parameters -stringentPaired for only high-quality fragments and with parameters -wgsMode for WGS data). The methylation level from WGBS was called by Bis-SNP v0.90[30].

**Non-homogeneous Hidden Markov Model.** The initiation matrix was summarized based on the methylation states of the first CpG in each DNA fragment separately (Fig. 2). A nonparametric model was used to calculate the initiation and transition matrix by considering the distance with adjacent CpG sites. A gaussian mixture model was applied to model the emission likelihood of each of the three fragmentation features (fragment length, coverage, and distance to the center of the fragment). A weighted DNA methylation prior, estimated from methylation level at genomic DNA (buffy coat) in healthy individuals, was utilized to calculate the posterior emission probability of hidden status only in the decoding (i.e., prediction) step, which models the base DNA methylation differences in different genomic contexts. For example, the probability of observing methylated event *em* given that located at the CpG site with methylation prior *k* is:

$$\Pr(e_m) = \frac{\Pr(e_m \mid k)\Pr(k)}{\Pr(e_m \mid k)\Pr(k) + \Pr(e_u \mid 1-k)(1-\Pr(k))} \tag{1}$$

Two states Hidden Markov Model (HMM) is implemented as described in Rabiner[31] at Jahmm framework with some adaptations to our problem. Baum-Welch algorithm was used to estimate the parameters with a maximum of 50 iterations. The model was trained by all the cfDNA fragments with at least 7 CpGs within the same fragments. The number of CpGs was not limited at the decoding step. In low-coverage data, we utilized an HMM model trained in high-coverage samples (HD_45, a healthy individual) to estimate the model parameters and applied it directly to each ULP-WGS dataset for the decoding. All details are implemented in FinaleMe.java (with parameters: -miniDataPoints 7 -gmm -covOutlier 3, for the training step and parameters -decodeModeOnly for the decoding step).

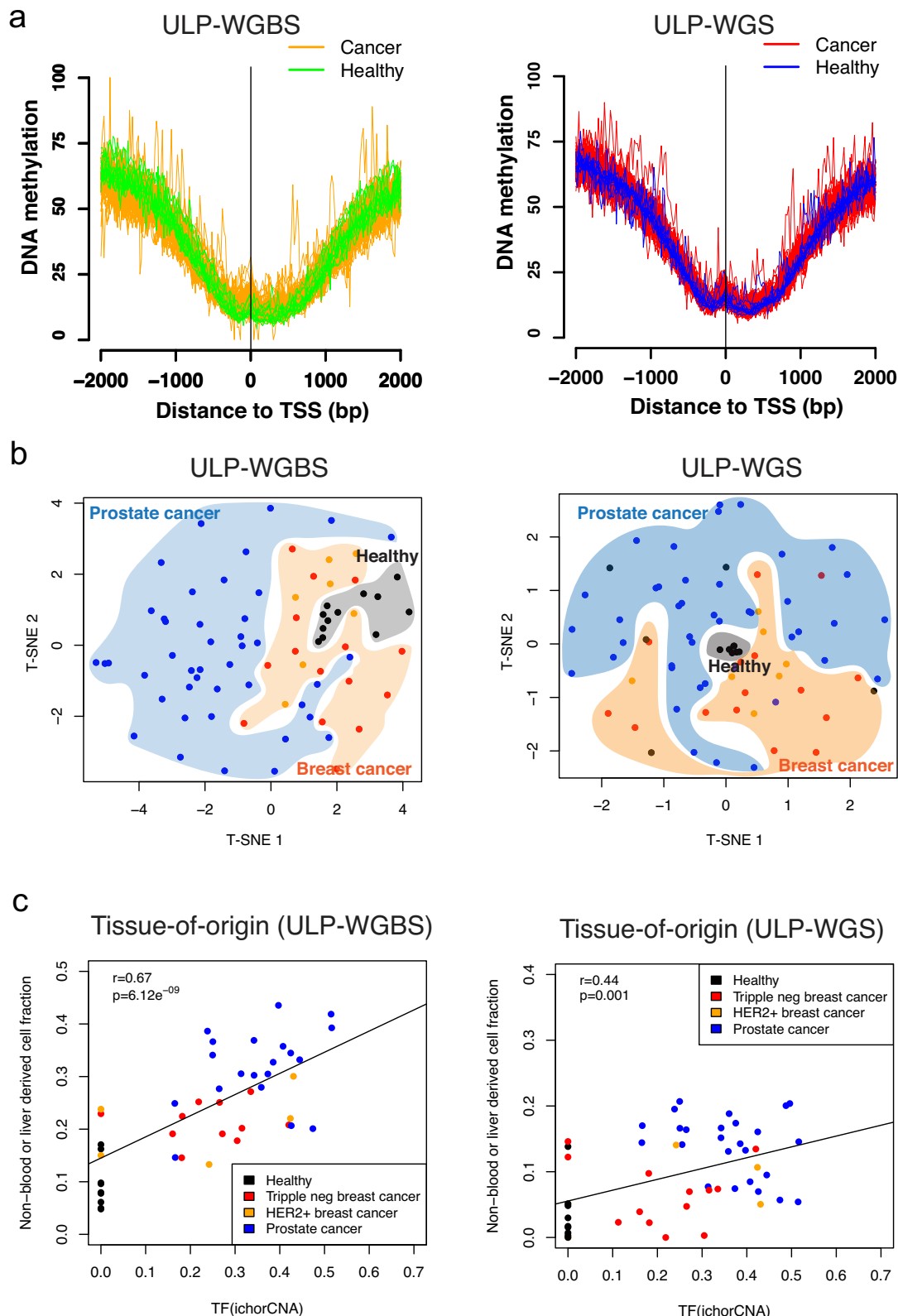

**Gaussian Mixture Model (GMM) initialization for HMM model.** GMM algorithm was utilized to estimate the initiation state of each CpG in each fragment by three fragmentation feature vectors with a maximum of 10,000 iterations. After GMM initialization, in WGBS, the methylated and unmethylated states were identified by the mean methylation level of each state. In WGS data, the state with a higher distance to the center was defined as the methylated state. Then the initiation

parameters of HMM model were estimated based on the GMM initialization.

**Initiation and transition probability.** The initiation probability of each state with the same offset from the start of the fragment was averaged by the states of the first CpGs with the same offset range at all the high-quality fragments. The transition probability matrix between states

**Fig. 4 | Inferring DNA methylation and tissues-of-origin from cfDNA ULP-WGS.**
**a** Average ground truth (ULP-WGBS, left panel) and predicted (ULP-WGS, right panel) DNA methylation level from cancer and healthy individuals at CpG island promoter regions ($n$ = 17,880). Orange lines represent the ground truths from ULP-WGBS in the cancer patients. Green lines represent the ground truths from ULP-WGBS in the healthy individuals. Red lines represent the predicted values from ULP-WGS in the cancer patients. Blue lines represent the predicted values from ULP-WGS in the healthy individuals. **b** T-SNE plot by using the DNA methylation level in the 100 kb non-overlapped window in autosomes but only summarized from CGI

and CGI shore regions in the ground truth (ULP-WGBS, left panel) and predicted (ULP-WGS, right panel) results from cancer (Orange: breast cancer, $n$ = 22. Blue: prostate cancer, $n$ = 43) and healthy individuals (Black, $n$ = 12). **c** the concordance of prostate or breast-related cell-type fractions (Ground truth from ULP-WGBS: left panel, Predicted from ULP-WGS: right panel) with tumor fraction estimated by ichorCNA in both healthy (Black, $n$ = 12) and cancers (Red: triple negative breast cancer, $n$ = 15. Orange: HER2+ breast cancer, n = 7. Blue: prostate cancer, $n$ = 43). The correlation coefficient and p-value are calculated by two.sided Pearson correlation test in cor.test function in R. Source data are provided as a Source Data file.

---

was also calculated separately for each of the possible distance ranges to the previous CpG.

**Emission distributions.** Three features were modeled by Multivariate Mixture Gaussian distribution. Two components mixture of Gaussian distribution was used to model each of the features separately.

$$\Pr\left(e_m \mid k\right) = (1 - \pi) \times N(\mu_i, \sigma_i^2) + \pi \times N\left(\mu_j, \sigma_j^2\right) \qquad (2)$$

In the Viterbi decoding step, methylation prior estimated from genomic DNA in buffy coat samples from healthy individuals[7] was only used to calculate the emission probability for each CpG.

**KL divergence.** Kullback-Leibler distance was used to estimate the divergence of new HMM during Baum-Welch re-estimation. Since methylation prior was used for the decoding step and is different at different CpG site, 10,000 random fragments with a minimum of 5 CpGs is selected to calculate the Kullback-Leibler distance. If the distance between new and old HMM was less than $1e^{-4}$ or the changes of distance were less than 1%, the model was considered converged.

**Summary of the model.** In cfDNA WGS (Fig. 2), our HMM model infers the model parameters directly from WGS data without using cfDNA WGBS data. The principle of the model is: we assume that there are two binary states (u or m) in each CpG at each cfDNA fragment. These two states are not observable in WGS (thus hidden). We assume that the states are affected by three fragmentation features. At each CpG in each fragment in the bam file (CpG point), we can obtain three features: the fragment's length, the CpG's distance to the center of that fragment, and the fragment coverage at that particular CpG position in the reference genome. We also assume the status of each CpG in each fragment is a Multivariate Gaussian distribution of these three features.

Step 1, we utilized a Gaussian mixture model to classify all the CpG points in WGS into two groups (u or m) to initiate the HMM model (the initial parameters). Given the hypothesis in Fig. 1B, we always assume "m" group has a larger average distance to the center of fragments.

Step 2, we applied the initiated parameters to the HMM model and built a Markov chain for each single cfDNA fragment. Due to the Markov process, the status of each CpG point is affected by its adjacent CpG in the same fragment. Then, the Baum-Welch algorithm was used to estimate the maximum likelihood parameters in the WGS dataset. Different from the traditional HMM model that assumes equal transition probability between CpGs, we utilized a non-homogenous model to estimate different transition probability matrices given different distances between CpGs. Kullback-Leibler distance was utilized to estimate whether or not the model converged during the iteration.

Step 3, after the estimation of parameters in step 2 (training), we utilize the Viterbi algorithm to estimate the best state (u or m) in each CpG at each fragment. Different from the traditional HMM model, we add methylation prior from WGBS in a healthy buffy coat to calculate the posterior probability.

Step 4, after the prediction in step 3, we aggregated the methylation status across fragments at each CpG site in the reference genome and calculated the continuous methylation level (0-100%).

**Performance evaluation**
**Comparison of the binary methylation status of each CpG in each fragment (WGBS).** The equal number of methylated and unmethylated CpGs was randomly sampled at the evaluation step. Prediction results were compared with ground truth methylation binary states at each CpG in each cfDNA fragment of WGBS. The threshold was varied to identify methylated status at the Viterbi decoding step in order to calculate the ROC curve.

**Comparison of the continuous methylation level at each CpG or windows in the reference genome (paired WGBS and WGS).** FinaleMe was trained and decoded at WGS data only. The methylation level was calculated by aggregating the binary methylation status across fragments at each CpG in the reference genome. Finally, the continuous methylation level at each CpG or window was compared with the methylation level obtained from matched WGBS in the same blood draw.

**Comparison of methylation profiles at important regulatory elements (paired WGBS and WGS).** FinaleMe was trained and decoded at WGS data. The predicted methylation level was calculated as described in above (section of Non-homogeneous Hidden Markov Model). The average methylation level around CpG island promoters, 5′ end of exons, and CTCF motifs were calculated by Bis-Tools as described in Lay & Liu et al.[32]. CpG island definition was downloaded from UCSC genome browser[33]. CpG island shore was defined by the regions within 2 kb regions around the CGI.

**Benchmark of the speed.** We downsampled the high-coverage cfDNA WGS data and calculated the time cost with different numbers of fragments in the bam files (Supplementary Fig. 8). Benchmark was performed at a single CPU in the computational cluster (Intel(R) Xeon(R) Gold 6338 CPU @ 2.0 GHz).

**Tissue-of-origin deconvolution.** To infer tissue of origin from measured or inferred DNA methylation data, we modeled patient methylation data as a linear combination of reference methylomes. We constrain the weights to sum up to one so that the weights can be interpreted as tissue contribution to cfDNA. Quadratic programming was utilized to solve the constrained optimization problem. This method and approach closely follow the tissue deconvolution algorithm described in Sun et al. PNAS[6]. To reduce the noise, we utilized the methylation density at 1 kb non-overlapped windows within the CpG island and CpG island shore regions at autosomes and binarized the methylation level (window with methylation density <0.1 was defined as 0, otherwise 1) in both reference methylomes and cfDNA data. Only windows with at least 10 Cs or Ts across all the reference methylomes were utilized for the analysis. Only windows that were highly variable across reference methylomes (top 1% most variable regions in the reference methylomes) were further utilized for the deconvolution.

We incorporated WGBS from the major immune cell types (Neutrophil, B cell, T cell, Macrophage, Erythroblast cells), blood vessel endothelial cells, and liver hepatocyte cells, as suggested by Moss 2018 Nature Communications[3]. We also incorporated methylomes from mammary epithelial cells (HMEC) and prostate epithelial cells (PrEC) since they are related to the cancer types we analyzed.

In the low pass data, we further relaxed our criteria about the coverage to keep more windows. The top 25% of most variable regions in the reference methylomes were utilized for deconvolution. Windows with less than 5 Cs or Ts in either reference methylome or cfDNA data were marked as NA. Samples or windows with more than 80% NA were filtered. We further imputed the missing data of the windows by K-nearest neighbor ($k = 5$ and maxp = "p" in impute.knn function at impute package, R 4.2.1) and finally binarized the methylation level within the window as that in high-coverage data.

**ichorCNA analysis.** Estimation of tumor fraction was performed using ichorCNA as described previously in Adalsteinsson et al. Nature Communications 2017[25]. Specifically, we utilized readCounter with parameters: --window 1000000 --quality 20 --chromosome "1,2,3,4,5,6,7,8,9,10,11,12,13,14,15,16,17,18,19,20,21,22,X,Y" to generate the wig files. Then we utilized runIchorCNA.R with parameters: --normal "c(0.75)" --scStates "c(1,3)" --ploidy "c(2)" --maxCN 5 together with gc_hg19_1000kb.wig, map_hg19_1000kb.wig, GRCh37.p13_centromere_UCSC-gapTable.txt, and HD_ULP_PoN_1Mb_median_normAutosome_mapScoreFiltered_median.rds panel provided by ichorCNA to calculate tumor fraction for each sample.

**Differential methylation analysis.** Differential methylation regions (predefined non-overlapped 1 kb windows in autosomes) in high-coverage WGBS were identified by metilene (v 0.2–8)[34] with $q$ value < 0.05. Data in ULP-WGBS are very sparse and noisy. Therefore, we utilized two-sided Wilcoxon Rank Sum Tests to identify the windows that were different between cancers and healthy controls with a $p$ value cut-off 0.01.

### Statistics and reproducibility
No statistical method was used to predetermine sample size. No data were excluded from the analyses. The experiments were randomized to generate cfDNA sequencing libraries. The Investigators were not blinded to allocation during experiments and outcome assessment.

### Reporting summary
Further information on research design is available in the Nature Portfolio Reporting Summary linked to this article.

## Data availability
The publicly available cfDNA WGBS data used in this study are available in the dbGaP database under accession code [https://www.ncbi.nlm.nih.gov/projects/gap/cgi-bin/study.cgi?study_id=phs000846.v1.p1][7]. The publicly available ULP-WGS data used in this study are available in the dbGaP database under accession code [https://www.ncbi.nlm.nih.gov/projects/gap/cgi-bin/study.cgi?study_id=phs001417.v1.p1][25]. The raw sequencing data for the deep WGS, WGBS. and ULP-WGBS data generated in this study have been deposited in the Sequence Read Archive with controlled access from dbGaP under accession code phs003287.v1.p1 [https://www.ncbi.nlm.nih.gov/projects/gap/cgi-bin/study.cgi?study_id=phs003287.v1.p1]. These data are available under restricted access due to individual privacy concerns. Permanent employees of an institution at a level equivalent to a tenure-track professor or senior scientist with laboratory administration and oversight responsibilities may request access through dbGaP. The requests, which are managed by NHGRI's Data Access Committee, take less than one month for approval, and access is permitted for 12 months. The processed and de-identified data are available in zenodo.org (https://doi.org/10.5281/zenodo.7779198)[35]. The remaining data are available within the Article, Supplementary Information, and Source Data file. Source data are provided with this paper.

## Code availability
Code for FinaleMe and associated scripts are publicly available on GitHub under the MIT license for academic researchers: https://github.com/epifluidlab/FinaleMe.git[36]. The zipped code is also available in zenodo.org (https://doi.org/10.5281/zenodo.7779198)[35].

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

## Acknowledgements

This work was supported by the computational resources from the Broad Institute of MIT and Harvard, the Biomedical Informatics (BMI) high-performance computing cluster in CCHMC, and QUEST computational cluster in Northwestern University. This work also used the Extreme Science and Engineering Discovery Environment (XSEDE), which is supported by the National Science Foundation grant number ACI-1548562. This work used the XSEDE at the Pittsburgh Supercomputing Center (PSC) through allocation MCB190124P and MCB190006P. Y.L. is supported by the Broad Next10 grant from the Broad Institute of MIT and Harvard, trustee award from Cincinnati Children's Hospital Medical Center, the startup grant to Y.L. from Cincinnati Children's Hospital Medical Center, Northwestern University, Robert H. Lurie Comprehensive Cancer Center of Northwestern University, and NHGRI (R56HG012360 to Y.L.). The authors acknowledge the generous support of the Gerstner Family Foundation to V.A.A., the Wong Family Foundation and DFCI Medical Oncology grant to A.D.C.

## Author contributions

Y.L., V.A.A. and M.K. conceived the study. Y.L. implemented the computational method. S.R. performed the library constructions. Y.L., C.L., D.W.K, R.B, G.H., G.G, J.R, D.R, H.Z., H.F. and S.F. performed the data analysis with input from A.D.C., H.A.P, D.G.S, V.A.A. and M.K. A.D.C., H.A.P. and D.G.S. provided the clinical samples and guidance related to the clinic applications. Y.L. and V.A.A. wrote the manuscript together. All authors read and approved the final manuscript.

## Competing interests

Y.L., V.A.A. and M.K. have an approved patent covered FinaleMe ("Methods for genome characterization", US Patent US11788135B2, date of patent, Oct 17, 2023, filed by MIT and Broad Institute of MIT and Harvard). Y.L. owns stocks from Freenome Inc. V.A.A., G.H. and S.F. are inventors on an approved patent covered ichorCNA (US20190078232A1, "Methods for genome characterization", date of patent, Mar 14, 2019, filed by Harvard College, Dana Farber Cancer Institute Inc, Broad Institute of MIT and Harvard) on methods for estimating tumor fraction in cfDNA. VAA is a co-inventor on a patent application covering MAESTRO (US 2023/0203568, "Minor allele enrichment sequencing through recognition oligonucleotides", pending, filed by Broad Institute of MIT and Harvard), which has been licensed to Exact Sciences, receives sponsored research funding from Exact Sciences, and is a cofounder and advisor to Amplifyer. The remaining authors declare no competing interests. H.Z is currently an employee at Regeneron Pharmaceuticals Inc. and contributed to this article as an employee of Cincinnati Children's Hospital Medical Center, and the views expressed do not necessarily represent the views of Regeneron Pharmaceuticals Inc.
