## [Peer Review File · Nature Communications]

FinaleMe: Predicting DNA methylation by the fragmentation patterns of plasma cell-free DNAReviewers' Comments:

Reviewer #1:

Remarks to the Author:

In this paper, the authors develop FinaleMe to predict DNA methylation of cfDNA and tissue-of-origin directly from plasma WGS. Their results are validated on 80 pairs of WGS and WGBS data. As cfDNA methylation analysis has many clinical utilities but needs specialized protocols and high input amounts, FinaleMe addresses an unfilled need to use the millions of cfDNA samples that have already been or will be profiled by genomic sequencing. The proposed HMM model can predict the methylation status at each individual CpG site in a DNA fragment. Previous studies used nucleosome spacing in WGS to do tissue of origin analysis by ranking the tissues; here the inferred methylation provides absolute fractions of each cell type. Tissue of origin profiles between predicted and measured methylation levels were consistent. Results were consistent with ULP-WGS. DMRs were consistent between WGS and WGBS. Overall, the paper is well-organized and the proposed method is solid, and can significantly increase the utilities of the cfDNA WGS data. The following critics shall help to improve the paper:

Figure 1a:

In Figure 1a, it is very interesting that there is a dip at 250bp. Would you be able to offer some insights about why?

Figure 1b:

Your diagram leads me to believe the cfDNA portion wrapped around the nucleosome is unmethylated, while the linker region is methylated. Is this true?

Of the three features used (fragment length, normalized coverage, and distance of each CpG to the center of the fragment), which is the most important? What happens if you use these features individually?

Supplementary Figure 1:

Your graph of jagged end fill in is opposite the graph presented in figure 1 in Jiang et al. Genome Research 2020 (reference 27). The drop off in methylation should occur in the beginning cycles of R2 (i.e., in the first ~40 bases of the R2 fastq files, since fastq files are read 5' -> 3'), when combined with R1, the beginning of the R2 fastq file is the 3' end of the original fragment. Can you clarify why your graph shows the drop off at the later cycles in R2, instead of in the initial cycles?

In the Data preprocessing section in methods, what do you mean by "2nd end of paired reads" and "1st end of paired reads"? From my understanding of jagged ends, it should affect the 5' end of R2, which is the 3' end of the original fragment.

In the original double-stranded cfDNA fragments, there are two strands, the original top (OT) and original bottom (OB). The fill in with unmethylated C during end repair occurs at the 3' end of the OT and OB strands.

During PCR, these two strands are separated. When the OT strand is sequenced, its second strand is complementary to the OT (CTOT), and is not the same as the OB strand due to bisulfite conversion. Because the OT had fill in at its 3' end, the CTOT strand has fill in at its 5' end. R1 sequences the first 1-100 bp of OT (5' -> 3'), whereas R2 sequences the first 100 bp of CTOT (5' -> 3'). This means the jagged end fills in the 3' end of R1 (likely not seen in sequencing data since original fragment length >100bp) and the 5' end of R2 (the beginning of R2, since fastq files are read 5'->3'). To remove the biased methylation information, the beginning (5' end) of R2 should be trimmed.

Methods, line 213:

Reads were trimmed to account for jagged end fill in with unmethylated cytosines. However, as described above, trimming the jagged ends alters the fragment length. It is not clear from your description if this is what you're doing, however. I think you may be trimming the wrong end of the individual reads in the read pair based on Supplementary figure 1. Regardless, why not just mask the methylation status of CpGs at these ends, and use the entire fragment for length estimates? You could also use data from single strand kits (e.g. Swift) to compare, since these do not undergo end repair and won't be subject to the same issues. Furthermore, did you also trim WGS data similarly? Otherwise the lengths are offset (shorter in WGBS compared to WGS).

Methods, line 233:

The HMM model was trained on fragments with at least 7 CpGs. Since fragments containing more CpGs are more likely to be in CGI/regions with high CpG density, don't these fragments have different properties from fragments in low CpG density regions? Wouldn't it be better to include low-density CpG regions in your model training? As you note that your model performs worse in CpG-poor regions (Results line 131), could you train a separate model for cpg-poor regions?

Supplementary figure 7:

I am confused what do the Xs mean? What do the row/column names mean? What does the slant of the ovals mean? Are the samples that are negatively correlated cancer vs normal? The figure is supposed to be showing that downsampled deep sequenced samples have the same tissue-of-origin results as the original sample, so why are totally different samples being compared?

Minor critiques:

Results, line 113:

Sampling an "even" number of methylated and unmethylated CpGs. Do you mean equal number?

Supplementary Fig 1

Supplementary Fig. 1 is referenced after all the other supplementary figures, it should be renumbered.

Reviewer #2:

Remarks to the Author:

Cell-free DNA (cfDNA) isolated from peripheral blood largely originates from nucleosomes that make the DNA less accessible to degradation by endonucleases. As a consequence, the cell-free DNA (cfDNA) fragmentome can comprehensively represent both genomic, chromatin, and methylation characteristics that reflect how DNA is packaged in cells, and has the potential to identify a large number of tumor-derived changes in the circulation. Bisulfite sequencing, while widely used for profiling methylation more directly, requires larger amounts of input DNA that may be prohibitive in many liquid biopsy applications. Liu et al. describe a hidden Markov model (HMM) that predicts the methylation status at CpG's genome-wide from whole genome sequencing of cfDNA. They compare inference from the HMM to whole genome bisulfite sequencing of the same samples as ground truth. Additionally, the authors use the predictions from their HMM to identify the tissue of origin in a small pilot study of two cancers (breast, prostate, and healthy). As previous studies have established connections between cfDNA characteristics and methylation, the innovation in this paper is primarily the use of a different approach, here a HMM, to infer methylation. It is not clear that inferring the methylation status has any added advantage of distinguishing the tissue of origin compared to fragmentation features alone, which have been shown to broadly reflect chromatin structure (e.g., Cristiano et al., 2019). The HMM is described at a high level making it difficult to understand the implementation, necessary inputs, assumptions, and limitations, and the datasets evaluated are

limited in number of available samples.

Major comments:

- Comparison to Fragma (Zhou et al., 2022) and other approaches (if available) to infer methylation from WGS of cfDNA

- More detailed description of the HMM implementation and features. This seems to be limited to Fig1a. The schematic in Fig 1b is helpful, but not clear how this connects to the HMM. I do not understand the implementation of the HMM, the dimensionality of the feature vectors, and the inputs. It is not clear what is meant by training the HMM, or if the user would also need to train the HMM. Terminology like 'decoding' should be defined. Do not refer the reader to "CpGMultiMetricsStats.java" or any code for details. Fragment coverage was normalized by dividing the total number of high-quality reads in the bam file. Not clear if the fragment coverage is calculated per CpG site, per CpG island, or some other characteristic. Therefore, the dimensionality of the fragment coverage feature vector is also not clear. Z-scores of fragment lengths are another feature. I assume the dimensionality of this feature vector is the same as the fragment coverage feature vector, but not clear. What was used to center and scale the fragment length z-score? Is this a within- or between-sample centering and scaling? The third feature is 'distance to the center of the fragment'. Is this feature fragment-specific? Does this suggest that the HMM integrates both fragment-level and region-level features to infer CpG status at a site? Are all CpG sites in the entire autosome evaluated -- how long does this take? Is a single HMM fit treating the genome as one long feature vector, or is it parallelized by chromosome arm for example. How are the parameters for the Guassian Mixture Model and other settings determined? Are these determined independently for each sample? Is the user of FinaleMe required to specify any of these parameters? If so, how sensitive are the HMM results to choices for these parameters?

- Is the tissue of origin from this approach any more accurate than existing approaches that have used fragmentation-based features directly to identify tissue of origin (e.g., Cristiano et al., 2019 and Lo et al. papers).

- In tissue of origin analyses, how were the samples batched and processed? Provide as supplemental table DNA extraction date, date of genomic library preparation, date of sequencing, flow cell and lane.

We appreciate your time and both reviewers' feedback on the manuscript. We have incorporated all the suggestions from the reviewers and highlighted these changes within the revised manuscript.

Here are the point-by-point responses to the reviewers' comments and concerns.

Comments from Reviewer 1

Reviewer #1, expertise in cfDNA methylation, tissue of origin prediction and machine learning
(Remarks to the Author):

In this paper, the authors develop FinaleMe to predict DNA methylation of cfDNA and tissue-of-origin directly from plasma WGS. Their results are validated on 80 pairs of WGS and WGBS data. As cfDNA methylation analysis has many clinical utilities but needs specialized protocols and high input amounts, FinaleMe addresses an unfilled need to use the millions of cfDNA samples that have already been or will be profiled by genomic sequencing. The proposed HMM model can predict the methylation status at each individual CpG site in a DNA fragment. Previous studies used nucleosome spacing in WGS to do tissue of origin analysis by ranking the tissues; here the inferred methylation provides absolute fractions of each cell type. Tissue of origin profiles between predicted and measured methylation levels were consistent. Results were consistent with ULP-WGS. DMRs were consistent between WGS and WGBS. Overall, the paper is well-organized and the proposed method is solid, and can significantly increase the utilities of the cfDNA WGS data. The following critics shall help to improve the paper:

Our overall response: Thanks for the positive comments!

Figure 1a:

In Figure 1a, it is very interesting that there is a dip at 250bp. Would you be able to offer some insights about why?

Our response: Thanks for pointing out this interesting observation! We indeed noticed that. Our current explanation is that this might be the cfDNA fragments bound with a single nucleosome+linker+half nucleosome: $\sim 147+20+73 \approx 250$ bp. Franklin Pugh lab's previous study (Rhee et al. 2014 Cell, PMID: 25480300) "detect widespread subnucleosomal structures in dynamic chromatin, including what appear to be half nucleosomes consisting of one copy of each histone.". There are some follow-up studies on this observation (PMID: 30661750, 28902852), but they are mostly based on other model organisms, not humans.

Figure 1b:

Your diagram leads me to believe the cfDNA portion wrapped around the nucleosome is unmethylated, while the linker region is methylated. Is this true?

Our response: Yes, people have observed this in genomic DNA for a while. Our previous studies in NOME-seq (Kelly, Liu, et al. 2012 Genome Res, PMID: 22960375) have shown a negative correlation between DNA methylation (black line in the figure below, left panel) and nucleosome occupancy (green line in the figure below, left panel).

There are also some follow-up studies getting the same conclusion (Huff et al. 2014 Cell, PMID: 24630728, figure below, right panel: grey and black is the nucleosome, dark blue is DNA methylation; Collings et al. 2017 Epigenetics Chromatin, PMID: 28413449)

Kelly et al. 2012 Genome Res

Huff et al. 2014 Cell

Of the three features used (fragment length, normalized coverage, and distance of each CpG to the center of the fragment), which is the most important? What happens if you use these features individually?

Our response: Thanks for the suggestion. First, in cfDNA WGBS data, we tried to just use single features for the training. We then decode it in CGI+shore regions. When comparing the ground truth methylation value in each read, we can see that “Distance to Center” contribute least in the current model and dataset (see Figure below). However, this model performance is evaluated based on 100PE WGBS data. After the trimming of jagged end from the reads, there are not a lot of data points with methylation values in the center of fragments, which may bias our evaluation here.

So we repeated the same step for cfDNA WGS and then compared the predicted methylation level result with the paired cfDNA WGBS at each CpG site (CpG island + shore regions). We can see that “distance to the fragment center” becomes the 2nd important feature in terms of the correlation at a single CpG site within CGI+shore regions (see Figure below).

Supplementary Figure 1:

Your graph of jagged end fill in is opposite the graph presented in figure 1 in Jiang et al. Genome Research 2020 (reference 27). The drop off in methylation should occur in the beginning cycles of R2 (i.e., in the first ~40 bases of the R2 fastq files, since fastq files are read 5' -> 3'), when combined with R1, the beginning of the R2 fastq file is the 3' end of the original fragment. Can you clarify why your graph shows the drop off at the later cycles in R2, instead of in the initial cycles?

Our response: Apologies for the mistake here. In Supplementary Figure 1a-b, we tried to replicate Figure 2B in Jiang et al.'s 2020 Genome Res paper (see Figure below) but did not realize the misleading information here. We have corrected the original figure (new Supplementary Figure 7) and added the reads orientation illustration there.

Jiang et al. 2020 Genome Res

Our manuscript

In the Data preprocessing section in methods, what do you mean by “2nd end of paired reads” and “1st end of paired reads”? From my understanding of jagged ends, it should affect the 5’ end of R2, which is the 3’ end of the original fragment.

Our response: Apologies for confusing the reviewer here. 2nd end of paired reads is R2 and 1st end of paired reads is R1. We have updated all the relevant descriptions in the manuscript.

In the original double-stranded cfDNA fragments, there are two strands, the original top (OT) and original bottom (OB). The fill in with unmethylated C during end repair occurs at the 3’ end of the OT and OB strands.

During PCR, these two strands are separated. When the OT strand is sequenced, its second strand is complementary to the OT (CTOT), and is not the same as the OB strand due to bisulfite conversion. Because the OT had fill in at its 3’ end, the CTOT strand has fill in at its 5’ end. R1 sequences the first 1-100 bp of OT (5’ -> 3’), whereas R2 sequences the first 100 bp of CTOT (5’ -> 3’). This means the jagged end fills in the 3’ end of R1 (likely not seen in sequencing data since original fragment length >100bp) and the 5’ end of R2 (the beginning of R2, since fastq files are read 5’->3’). To remove the biased methylation information, the beginning (5’ end) of R2 should be trimmed.

Our response: Apologies for the mistake and thanks very much for pointing it out. We trimmed the 5' end of R2, regenerated all related figures and analysis, and the results are still largely the same.

Methods, line 213:

Reads were trimmed to account for jagged end fill in with unmethylated cytosines. However, as described above, trimming the jagged ends alters the fragment length. It is not clear from your description if this is what you're doing, however. I think you may be trimming the wrong end of the individual reads in the read pair based on Supplementary figure 1. Regardless, why not just mask the methylation status of CpGs at these ends, and use the entire fragment for length estimates?

Our response: Thanks for the great suggestion! In the revised manuscript, after the adapter trimming, we masked the methylation status in cfDNA WGBS (For R2, we masked methylation status at CpG in the first 40bp at 5' end and the last 10bp at 3' end. For R1, we masked methylation status at CpG in the first 10bp at 5' end and last 40bp at 3' end). The conclusion related to cfDNA WGBS (Figure 1) remains the same.

You could also use data from single strand kits (e.g. Swift) to compare, since these do not undergo end repair and won't be subject to the same issues. Furthermore, did you also trim WGS data similarly? Otherwise the lengths are offset (shorter in WGBS compared to WGS).

Our response: Thanks for the suggestion. However, in Accel-NGS (swift single-strand protocol), the adaptase actually did the end-repair (please check the manual: <https://www.genetargetsolutions.com.au/wp-content/uploads/2015/05/Accel-NGS-Methyl-Seq-DNA-Library-Kit-Manual.pdf> and the figure below).

Protocol Overview

- The Accel-NGS Methyl-Seq protocol sequentially attaches adapters to single-stranded DNA fragments.
- The Adaptase step is a highly efficient, proprietary reaction that simultaneously performs end repair, tailing of 3' ends, and ligation of the first truncated adapter complement to 3' ends.
- The Extension step is used to incorporate truncated adapter 1 by a primer extension reaction.
- The Ligation step is used to add the second truncated adapter to the bottom strand only.
- The Indexing PCR step increases yield and incorporates full length adapters.
- Bead-based SPRI clean-ups are used to remove both oligonucleotides and small fragments, as well as to change enzymatic buffer composition.

There are also other resources about this protocol:

[https://www.bioscience.co.uk/userfiles/pdf/PRT-019%20Methyl-Seq%20Protocol%20Rev%203%20\(1\).pdf](https://www.bioscience.co.uk/userfiles/pdf/PRT-019%20Methyl-Seq%20Protocol%20Rev%203%20(1).pdf)

“The Accel-NGS Methyl-Seq Kit adds bases to 3’ termini during the Adaptase tailing step, including unmethylated cytosines. This tail adds a synthetic sequence, adding methylation information to the dataset. Therefore, trimming is required for Accel-NGS Methyl-Seq libraries to obtain improved mapping efficiency (with tools like Bismark or BSMAP) and precise methylation information and bisulfite conversion efficiency. Many informatics pipelines already include trimming of up to 10 bases from the beginning of both R1 and R2 to eliminate any synthetic cytosine methylation introduced as a result of filling in overhangs during end repair steps of conventional dsDNA library preparation and low quality bases due to bisulfite treatment.”

We did not trim WGS as the way in WGBS. In WGS data, we only trimmed the adapter when the adapter contamination was detected in some of the reads.

Methods, line 233:

The HMM model was trained on fragments with at least 7 CpGs. Since fragments containing more CpGs are more likely to be in CGI/regions with high CpG density, don't these fragments have different properties from fragments in low CpG density regions? Wouldn't it be better to include low-density CpG regions in your model training? As you note that your model performs worse in CpG-poor regions (Results line 131), could you train a separate model for cpg-poor regions?

Our response: Thanks for the suggestion. HMM model takes advantage of information from adjacent CpG methylation status within the same fragment. Due to the short cfDNA fragment, the power is significantly reduced in CpG-poor regions (1 CpG in a single cfDNA fragment, ~167bp). We indeed tried to train the current model in the CpG-poor region. With only three summary statistical features, we could not achieve high performance.

Generating a model that performs well in CpG-poor regions requires a more complicated model. Currently, we are working on a deep learning model, which will bring more information from the base composition in cfDNA fragment and reference genome, to enhance the performance in CpG-poor regions. However, that is beyond the scope of this paper.

Supplementary figure 7:

I am confused what do the Xs mean? What do the row/column names mean? What does the slant of the ovals mean? Are the samples that are negatively correlated cancer vs normal? The figure is supposed to be showing that downsampled deep sequenced samples have the same tissue-of-origin results as the original sample, so why are totally different samples being compared?

Our response: Apologies for the confusion. We would like to show that tissues-of-origin is highly correlated between high-coverage and downsampled data. The percentage of tissues that contributed to cfDNA was first calculated in each sample. Then the correlation between these tissues-of-origin vectors was calculated and compared between high-coverage and downsampled data. "corrplot" package in R was utilized to visualize the correlation. "X" on top of the plot means that "the correlation is not statistically significant ($p > 0.05$)". The shape of the plot represents the dispersion status of the dot.

For the row and column names, we apologize for using the original sample label and forgetting to make it consistent with other figures. We should not compare different samples here. We have already corrected it in the new Supplementary Figure 6 with updated figure legend (also attached here).

Supplementary Figure 6

Minor critiques:

Results, line 113:

Sampling an “even” number of methylated and unmethylated CpGs. Do you mean equal number?

Our response: Sorry about the mistake here. It should be “equal”. We have corrected it in the manuscript.

Supplementary Fig 1

Supplementary Fig. 1 is referenced after all the other supplementary figures, it should be renumbered.

Our response: Sorry about the mistake. We have corrected it in the manuscript.

Comments from Reviewer 2

Reviewer #2, expertise in cfDNA fragmentation patterns (Remarks to the Author):

Cell-free DNA (cfDNA) isolated from peripheral blood largely originates from nucleosomes that make the DNA less accessible to degradation by endonucleases. As a consequence, the cell-free DNA (cfDNA) fragmentome can comprehensively represent both genomic, chromatin, and methylation characteristics that reflect how DNA is packaged in cells, and has the potential to identify a large number of tumor-derived changes in the circulation. Bisulfite sequencing, while widely used for profiling methylation more directly, requires larger amounts of input DNA that may be prohibitive in many liquid biopsy applications. Liu et al. describe a hidden Markov model (HMM) that predicts the methylation status at CpG's genome-wide from whole genome sequencing of cfDNA. They compare inference from the HMM to whole genome bisulfite sequencing of the same samples as ground truth. Additionally, the authors use the predictions from their HMM to identify the tissue of origin in a small

pilot study of two cancers (breast, prostate, and healthy). As previous studies have established connections between cfDNA characteristics and methylation, the innovation in this paper is primarily the use of a different approach, here a HMM, to infer methylation. It is not clear that inferring the methylation status has any added advantage of distinguishing the tissue of origin compared to fragmentation features alone, which have been shown to broadly reflect chromatin structure (e.g., Cristiano et al., 2019). The HMM is described at a high level making it difficult to understand the implementation, necessary inputs, assumptions, and limitations, and the datasets evaluated are limited in number of available samples.

Our overall response: Thanks for the comments and suggestions! The previous work has pointed out the relationship between DNA methylation and cfDNA fragmentation (Jensen et al. 2015 Genome Biol, PMID: 25886572; Zhou et al. 2022 PNAS, PMID: 36288287; An et al. 2023 Nature Communications, PMID: 36653380). The previous approach can infer the DNA methylation status by using cfDNA WGBS (Zhou et al. 2022). However, their deep learning approach requires training on the ultra-deep coverage (merged 21 samples with >50X coverage) cfDNA WGBS dataset, and then possibly can be applied to cfDNA WGS (which is not shown in the paper!). WGBS and WGS are two completely different datasets. For example, bisulfite treatment causes sequence degradation in a sequence-dependent way (Grunau et al. 2001 NAR, PMID: 11433041), which affects the fragment length in different genomic regions. More specifically to cfDNA, end repair during the library construction in WGBS will bring the artificially low methylation at the ends of fragments (named jagged ends, Jiang et al. 2020 Genome Res). We could imagine that developing a deep learning model trained in cfDNA WGBS to WGS would require more extensive training datasets (after excluding the jagged end effect and bisulfite effect) and tuning of the model structure/parameters.

By contrast, our model can be applied to cfDNA WGS directly, without using cfDNA WGBS data for training. Moreover, as reviewer #1 noted, our model can predict the methylation status of each CpG in each cfDNA fragment. Therefore, our model can be trained and predicted not only on the same high coverage (16-39X) WGS data but also from ultra-low-pass WGS data (~0.1X), which could have wider clinical applications.

In cfDNA WGS (new Figure 2 shown above), our HMM model infers the model parameters directly from WGS data without using cfDNA WGBS data. The principle of the model is: we assume that there are two binary states (“u” or “m”) in each CpG at each cfDNA fragment. These two states are not observable in WGS (thus “hidden”). We assume that the states are affected by three fragmentation features. At each CpG in each fragment in the bam file (“CpG point”), we can obtain three features: the fragment’s length, the CpG’s distance to the center of that fragment, and the fragment coverage at that particular CpG position in the reference genome. We also assume the status of each CpG in each fragment is a mixed Multivariate Gaussian distribution of these three features.

Step 1, we utilized a Gaussian mixture model to classify all the “CpG points” in WGS into two groups (“u” or “m”) to initiate the HMM model (the initial parameters). Given the hypothesis in Figure 1B, we always assume “m” group has a larger average distance to the center of fragments (nucleosome linker are methylated, supported by previous studies: PMID: 22960375, PMID: 24630728, PMID: 28413449).

Step 2, we applied the initiated parameters to the HMM model and built a Markov chain for each single cfDNA fragment. Due to the Markov process, the status of each “CpG point” is affected by its adjacent CpG in the same fragment. Then, the Baum-Welch algorithm was used to estimate the maximum likelihood parameters in the WGS dataset. Different from the

traditional HMM model that assumes equal transition probability between CpGs in the same fragment, we utilized a non-homogenous model to estimate different transition probability matrices given different distances between CpGs. Kullback-Leibler distance was utilized to estimate whether or not the model converged during the iteration.

Step 3, after the estimation of parameters in step 2 (training), we utilize the Viterbi algorithm to estimate the best state (“u” or “m”) in each CpG at each fragment. Different from the traditional HMM model, we add methylation prior from WGBS in a healthy buffy coat to calculate the posterior probability.

Step 4, after the prediction in step 3, we aggregated the methylation status across fragments at each CpG site in the reference genome and calculated the continuous methylation level (0-100%) for each CpG in the reference genome.

In terms of novelty, we are the first to show that by using cfDNA WGS alone, we can predict the methylation status for each CpG within CpG-rich regions. Utilizing other published tissue-of-origin algorithms for cfDNA methylation, we can estimate the accurate percentage of cell type from each reference methylome. However, the other fragmentomic feature studies (Cristiano et al. 2019 Nature and Snyder et al. 2016 Cell) only gave the relative estimation that the sample is mostly likely from a particular disease or category (not the cell type).

This is also pointed out by Reviewer #1: “Previous studies used nucleosome spacing in WGS to do tissue of origin analysis by ranking the tissues; here the inferred methylation provides absolute fractions of each cell type...”

We apologize for not explaining the model in more detail in the manuscript. Now, we have incorporated a more complete schematic of our method in Fig 2 as well as more detailed information about it in the Methods section.

Major comments:

- Comparison to Fragma (Zhou et al., 2022) and other approaches (if available) to infer methylation from WGS of cfDNA

Our response: The FRAGMA source code was not released in the Zhou et al. 2022 paper. We contacted the corresponding authors after we obtained the reviewer’s comments (2023 Jun). However, after half a year’s effort (2023 Dec), we were unable to obtain the MTA/DTA between the two organizations and, therefore, could not obtain the code, model, and raw data for academic usage. Currently, there are no other methods available for the benchmarking.

Therefore, we reimplemented FRAGMA on our own based on the description from Zhou et al. 2022 (our implementation is available here: <https://github.com/epifluidlab/FRAGMA>). We obtained the features from our own cfDNA WGBS data in the manuscript and confirmed the differences in cleavage patterns between methylated and unmethylated CpGs ((following the instructions in the Methods section on how to filter CpG, see figure below, centered by CpG).

We trained the model on these cfDNA WGBS data and made the prediction of binary CpG methylation status in both paired cfDNA WGBS and WGS data. The performance of the model does not work well in our dataset in either whole genome or CpG-rich regions (see figure below), possibly due to the relatively low coverage (~20-30X, even after merging the healthy WGBS datasets) of the WGBS.

- More detailed description of the HMM implementation and features. This seems to be limited to Fig1a. The schematic in Fig 1b is helpful, but not clear how this connects to the HMM. I do not understand the implementation of the HMM, the dimensionality of the feature vectors, and the inputs. It is not clear what is meant by training the HMM, or if the user would also need to train the HMM. Terminology like 'decoding' should be defined. Do not refer the reader to "CpGMultiMetricsStats.java" or any code for details.

Our response: Thanks for the suggestion. We have added a new Figure 2 to illustrate the whole structure of the model (shown below)

The HMM model will be trained on the cfDNA WGS itself and then learned to predict (i.e. decode) the methylation status at each CpG in each fragment. Finally, the continuous methylation level (0-100%) will be calculated on each CpG in the reference genome.

Fragment coverage was normalized by dividing the total number of high-quality reads in the bam file. Not clear if the fragment coverage is calculated per CpG site, per CpG island, or some other characteristic. Therefore, the dimensionality of the fragment coverage feature vector is also not clear.

Our response: Please refer to the figure above (and Figure 2 in the manuscript). In each CpG site in the reference genome, we will calculate the fragment coverage (the number of fragments overlapped with that CpG in the reference genome) and then normalize it by dividing it by the total number of high-quality fragments in the bam file. For each CpG in each DNA fragment (CpG point), there are three features: fragment coverage, fragment length, and its distance to the fragment center. Then, we transformed these three features to their Z-score by using the mean and standard deviation of these three features across all the CpG points within the same bam file. Finally, we input these CpG points into the model for training.

Z-scores of fragment lengths are another feature. I assume the dimensionality of this feature vector is the same as the fragment coverage feature vector, but not clear. What was used to center and scale the fragment length z-score? Is this a within- or between-sample centering and scaling?

Our response: For each CpG in each DNA fragment (CpG point), there are three features: fragment coverage, fragment length, and its distance to the fragment center. The three features (fragment length, normalized coverage, and distance to the center of the fragment) were transformed into a Z-score by the mean and standard deviation of the features within the same bam file as the input for the HMM model. For example, for the Z-score transformation of fragment length, we used the average fragment length and standard deviation of the fragment length in the whole bam file to calculate the Z-score.

The third feature is 'distance <of CpG> to the center of the fragment'. Is this feature fragment-specific?

Our response: No, it is specific for each CpG in each fragment (CpG point). Please refer to the figure above (and our new Figure 2 in the manuscript). We will calculate the distance of this CpG to the center of the fragment (that CpG belongs to).

Does this suggest that the HMM integrates both fragment-level and region-level features to infer CpG status at a site?

Our response: No. Please refer to the figure above (our new Figure 2 in the manuscript). We only infer the binary methylation status for each CpG in each fragment (CpG point). In the end, we will aggregate the methylation level as that in WGBS to calculate the continuous methylation level (0-100%) for each single CpG in the reference genome.

Are all CpG sites in the entire autosome evaluated -- how long does this take?

Our response: Yes, all CpG sites in the whole autosome will be utilized. For the time cost, we added a new supplementary figure 8, shown below. Basically, for a bam file with ~550 million high-quality fragments, the training and decoding cost is ~52 minutes for a single CPU in the computational cluster (Intel(R) Xeon(R) Gold 6338 CPU @ 2.0GHz).

**Training and Decoding Time
vs.
Number of Fragments**

Is a single HMM fit treating the genome as one long feature vector, or is it parallelized by chromosome arm for example.

Our response: We utilize each CpG in each fragment (CpG point) as an input feature. We treat each DNA fragment as a Markov chain (the length depends on the number of CpG in each fragment). This is why the model works very well in CpG-rich regions but not in CpG-poor regions. Then we feed all the fragments in the same bam file to train one single HMM model.

How are the parameters for the Gaussian Mixture Model and other settings determined?

Our response: The parameters were all estimated from each sample by themselves. In the first step, we use Gaussian mixture models (no Markov chain) to classify all the CpG points in WGS into two groups (“u” or “m”) to initiate the HMM model (the initial parameters). Given the hypothesis in Figure 1B, we always assume “m” group has a larger distance to the center of fragments.

In step 2, we applied the initiated parameters to the HMM model and built a Markov chain for each single cfDNA fragment. Due to the Markov process, each CpG point status is affected and only affected by its adjacent CpG in the same fragment. Then, the Baum-Welch algorithm was used to estimate the maximum likelihood parameters in the WGS dataset. Different from the traditional HMM model that assumes equal transition probability between CpGs, we utilized a non-homogenous model to estimate different transition probability matrices given different distances between CpGs. Kullback-Leibler distance is utilized to estimate whether or not the model is converged during the iteration.

Step 3, after the estimation of parameters in step 2 (training), we utilize the Viterbi algorithm to estimate the best state (“u” or “m”) in each CpG. Different from the traditional HMM model, we add methylation prior from WGBS in a healthy buffy coat to calculate the posterior probability.

Step 4, after the prediction in step 3, we aggregated the methylation status across fragments at each CpG site in the reference genome and calculated the continuous methylation level (0-100%).

Are these determined independently for each sample?

Our response: Yes. Parameters were obtained from each cfDNA WGS sample itself and utilized to predict the methylation status from its own bam file.

Is the user of FinaleMe required to specify any of these parameters? If so, how sensitive are the HMM results to choices for these parameters?

Our response: The only parameters we tuned are: in the training step, what is the minimum number of CpG within the same fragments that we should use for the training? This is related to the accuracy of the HMM model estimation due to the Markov chain characters. We tried a minimum of 3, 5, and 7 CpGs within a cfDNA fragment, and the results are largely similar.

- Is the tissue of origin from this approach any more accurate than existing approaches that have used fragmentation-based features directly to identify tissue of origin (e.g., Cristiano et al., 2019 and Lo et al. papers).

Our response: Since we can predict the continuous methylation status in each single CpG, we can obtain similar results as the direct measurement from WGBS rather than the binary status as that from Zhou et al. 2022 PNAS paper (CpG methylation <0.3 was classified as “u” and methylation >0.7 was classified as “m”). Therefore, we can adapt other published tissues-of-origin algorithms designed for cfDNA methylation (we adopt the algorithm from Sun et al. 2015 PNAS paper). We can estimate the accurate percentage of cell type from each reference methylome. However, the other fragmentomic studies (Cristiano et al. 2019 Nature and Snyder et al. 2016 Cell) only gave the relative estimation that the sample is mostly likely from a particular disease or category (not about the percentage of cell types).

This is also pointed out by Reviewer #1: “Previous studies used nucleosome spacing in WGS to do tissue of origin analysis by ranking the tissues; here the inferred methylation provides absolute fractions of each cell type...”

- In tissue of origin analyses, how were the samples batched and processed? Provide as supplemental table DNA extraction date, date of genomic library preparation, date of sequencing, flow cell and lane.

.Our response: We have added the batch information in Supplementary Tables 2-3 for each sample.

Reviewers' Comments:

Reviewer #1:

Remarks to the Author:

The authors have addressed all my concerns, and I am satisfied with the revised version.